# Immune signature of acute pharyngitis in a *Streptococcus pyogenes* human challenge trial

Jeremy Anderson[1,2], Samira Imran [1,2], Hannah R. Frost[1], Kristy I. Azzopardi[1], Sedigheh Jalali[1,2], Boris Novakovic[1,2], Joshua Osowicki [1,2,3,5✉], Andrew C. Steer [1,2,3,5✉], Paul V. Licciardi [1,2,5✉] & Daniel G. Pellicci [1,2,4,5✉]

*Streptococcus pyogenes* causes at least 750 million infections and more than 500,000 deaths each year. No vaccine is currently available for *S. pyogenes* and the use of human challenge models offer unique and exciting opportunities to interrogate the immune response to infectious diseases. Here, we use high-dimensional flow cytometric analysis and multiplex cytokine and chemokine assays to study serial blood and saliva samples collected during the early immune response in human participants following challenge with *S. pyogenes*. We find an immune signature of experimental human pharyngitis characterised by: 1) elevation of serum IL-1Ra, IL-6, IFN-γ, IP-10 and IL-18; 2) increases in peripheral blood innate dendritic cell and monocyte populations; 3) reduced circulation of B cells and CD4+ T cell subsets (Th1, Th17, Treg, TFH) during the acute phase; and 4) activation of unconventional T cell subsets, γδTCR + Vδ2+ T cells and MAIT cells. These findings demonstrate that *S. pyogenes* infection generates a robust early immune response, which may be important for host protection. Together, these data will help advance research to establish correlates of immune protection and focus the evaluation of vaccines.

[1] Murdoch Children's Research Institute, Melbourne, Australia. [2] Department of Paediatrics, University of Melbourne, Melbourne, Australia. [3] Infectious Diseases Unit, Department of General Medicine, The Royal Children's Hospital Melbourne, Melbourne, Victoria, Australia. [4] Department of Microbiology and Immunology, University of Melbourne, Melbourne, Australia. [5] These authors contributed equally: Joshua Osowicki, Andrew C. Steer, Paul V. Licciardi, Daniel G. Pellicci. ✉email: joshua.osowicki@rch.org.au; andrew.steer@rch.org.au; paul.licciardi@mcri.edu.au; dan.pellicci@mcri.edu.au

*S*treptococcus pyogenes is a human-restricted pathogen with a profound and persistent global burden of disease[1]. It is a ubiquitous human pathogen and is responsible for more than 500,000 deaths each year[2,3]. Acute pharyngitis is the most common of all clinical syndromes due to *S. pyogenes*[4]. The diverse clinical spectrum of *S. pyogenes* also includes severe diseases including scarlet fever, cellulitis, necrotising fasciitis, toxic shock syndrome and post-infectious glomerulonephritis, acute rheumatic fever, and its sequelae rheumatic heart disease[1]. No vaccine is currently available for *S. pyogenes* and greater efforts are required to develop new vaccines or therapeutics to prevent serious disease and death caused by this bacterium.

Critical knowledge gaps have hindered *S. pyogenes* vaccine development[5]. Understanding the human immune response to *S. pyogenes* and identifying correlates of protection could accelerate the development of effective vaccines[6]. Animal models and in vitro studies are limited in their capacity to replicate the full clinical spectrum of human diseases caused by *S. pyogenes*[7]. Previous experimental human challenge studies were carried out decades ago, prior to the sophisticated immunological technologies available today[8–10]. As well as providing a platform for evaluation of candidate vaccines, a human-challenge model of *S. pyogenes* infection offers the unique opportunity to interrogate the dynamic host immune response to infection using samples collected before, during, and after development of disease. Human-challenge models have been established for other infectious pathogens such as *Vibrio cholerae*, *Salmonella* Typhi and Paratyphi, *Streptococcus pneumoniae*, respiratory syncytial virus and malaria, and have enormous potential to successfully guide vaccine development[11–15]. Recently, a new human-challenge model of *S. pyogenes* pharyngitis in healthy adults was established in an initial dose-finding clinical trial, after an extensive effort to select and characterise a challenge strain and develop a study protocol to protect participants and promote the model's clinical relevance[16–18].

In this study, we undertook comprehensive immune profiling of the early human immune response in 25 healthy adults from this initial trial who were challenged with *S. pyogenes*, 19 of whom subsequently developed acute symptomatic pharyngitis[17]. We focused our studies on the acute immune response, prior to treating participants with antibiotics to cure disease. We used high-dimensional flow cytometry and serum and saliva cytokine and chemokine analyses to identify an immune signature associated with acute pharyngitis, that will aid in the development of a future vaccine for *S. pyogenes*.

## Results

**Human-challenge model of S. pyogenes pharyngitis**. As reported previously, 25 healthy adult participants (12♀, 13♂, mean age 27.6 years) were enrolled and challenged with *emm75 S. pyogenes* applied by swab directly to the oropharynx (Fig. 1)[17]. Of the 25 participants, 20 were challenged from single-dose vials containing $1.72 \times 10^5$ CFU of *emm75 S. pyogenes*, while the remaining 5 were challenged with $1.62 \times 10^4$ CFU. After challenge, 18 participants met pre-defined clinical and microbiologic criteria for *S. pyogenes* pharyngitis (P+). Subsequent review of clinical, microbiological and immunological results found a protocol deviation affecting an additional volunteer (participant 057) who was previously described as 'likely pharyngitis' in the previous paper and considered in the P+ group here[17]. Six participants did not develop pharyngitis (P−) (Fig. 1)[17]. Pharyngitis was generally diagnosed between 36 and 72 h (h) after challenge. All participants were treated with antibiotics, either at the time of diagnosis (P+) or ~120 h after challenge in those without pharyngitis. Pharyngeal colonisation by the challenge strain was measured by quantitative

PCR[19], and a cycle threshold (CT)-value >35 was classified as no pharyngeal colonisation with *S. pyogenes*. All 19 P+ participants reported CT-values indicative of pharyngeal colonisation (median 21.24 at the timepoint of pharyngitis diagnosis, IQR 20.61–22.64) while only one P− participant exhibited a CT-value of <35 at any timepoint[17].

**Distinct cytokine and chemokine profiles are associated with S. pyogenes pharyngitis**. We previously demonstrated that IFN-γ and IL-6 were increased in the serum of P+ compared to P− participants by 48 h[17]. To gain a greater understanding of the soluble factors involved in the acute immune response to *S. pyogenes* challenge, we used multiplex assays to assess a larger panel of cytokines and chemokines in the saliva and serum of participants. Following challenge, saliva IL-1β, IL-1Ra and IL-6 were all increased at 24 h and IL-18 was also increased at 72 h in P+ participants (Fig. 2a). Most other soluble factors in the saliva did not change over time in the P+ participants (Supplementary Fig. 1).

Analysis of the serum of P+ participants also revealed increases in IL-1Ra, IL-6, IFN-γ and IL-18, with the greatest differences observed at 72 h, consistent with a delay in the kinetics of the immune response from the site of infection to the blood (Fig. 2b, c). Notably, levels of the chemokine IP-10 (CXCL10) were also greatly increased following challenge with *S. pyogenes* (Fig. 2b, c). IP-10 can directly kill *S. pyogenes* which suggests elevated IP-10 may promote protective immunity by enhancing bacterial clearance[20]. In most cases, the levels of these soluble factors were significantly higher at 72 h in P+ compared to P− participants (Fig. 2b, c). A reduction in IL-1β, IL-4, IL-7, IL-8, IL-9, IL-17A, eotaxin, and MCP-1 was also observed at 72 h in the serum of P+ participants, while levels of RANTES, TNF-α and MIP-1β did not change (Fig. 2b and Supplementary 2). Unsupervised principal component analysis (PCA) confirmed the differences observed between P+ and P− participants at 72 h, with IP-10, IL-1Ra and IL-18 the largest contributors in the early immune response to *S. pyogenes* (Fig. 2d). Overall, these data identify changes in chemokine and cytokine expression associated with the development of acute pharyngitis.

**Cellular signature to experimental human S. pyogenes pharyngitis**. Flow cytometry was used to characterise 43 immune cell populations, including innate cells, T cells and B cells in peripheral blood mononuclear cells (PBMCs) (Supplementary Figs. 3–5). Innate cells, such as intermediate and non-classical monocytes increased over time in P+ participants but not in P− participants, and classical monocytes were significantly elevated in P+ participants at 72 h compared to P− participants (Fig. 3a). Total dendritic cells and myeloid dendritic cells increased over time and were higher in P+ participants compared to P− participants, while plasmacytoid dendritic cells in P+ participants increased at 24 h (Fig. 3b). A decrease in CD56[bright] natural killer (NK) cells occurred from 24 to 72 h, and a rise of perforin expression at 72 h was also observed, although no other major changes in NK cell populations were observed between P+ and P− participants (Supplementary Fig. 6a). In response to *S. pyogenes*, monocytes and dendritic cells are crucial to phagocytosis, pro-inflammatory cytokine secretion, neutrophil recruitment and T cell activation[7,21]. In mice, depletion of myeloid dendritic cells resulted in uncontrolled dissemination of *S. pyogenes*[22]. Taken together, our results suggest monocytes and dendritic cells play a key role in experimental human *S. pyogenes* pharyngitis.

B cells decreased in P+ participants suggesting migration to the site of infection. No changes were observed in memory B cells, transitional B cells and plasmablasts during acute disease (Fig. 3c),

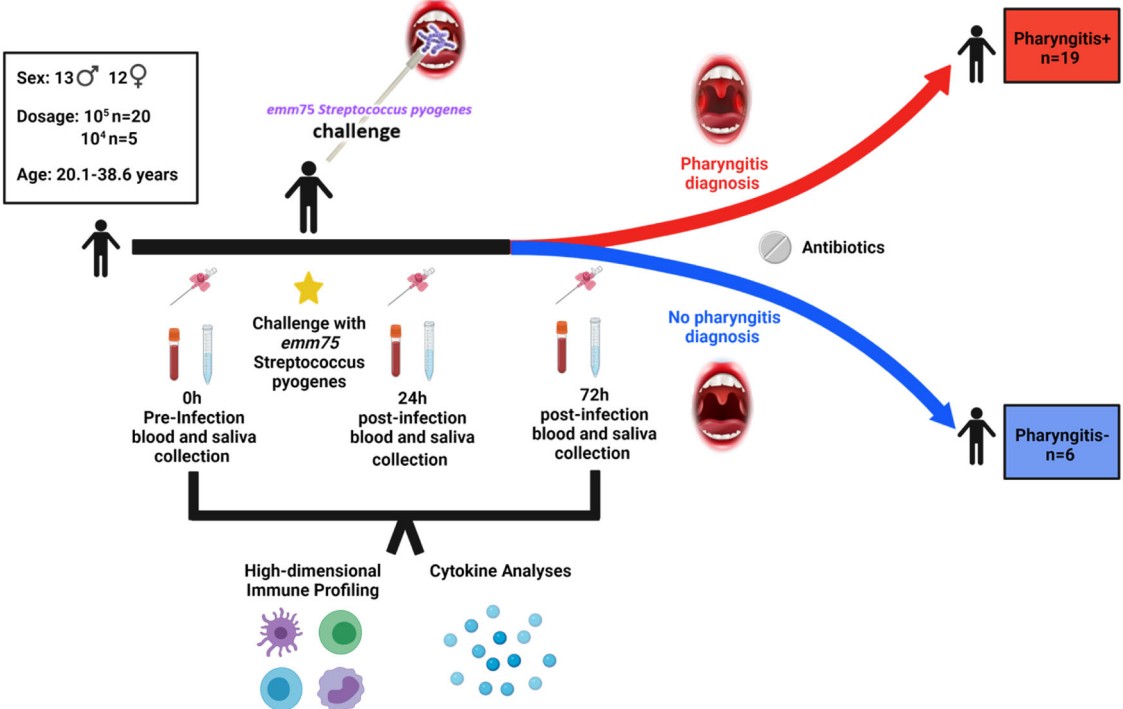

**Fig. 1 Human challenge with *Streptococcus pyogenes*.** Schematic of *Streptococcus pyogenes* human-challenge model. Details participant demographics, timepoints, outcomes and analyses performed. Figure created with BioRender.com.

and no differences were observed in the surface expression of IgD, IgM and IgG on B cells between P+ and P− participants (Supplementary Fig. 6b). Unconventional T cell populations such as mucosal-associated invariant T (MAIT) cells and γδTCR + Vδ2+ T cells are activated during infection by other respiratory pathogens and bridge the gap between innate and adaptive immunity[23]. The proportion of MAIT cells was similar between P+ and P− participants, however, γδTCR + Vδ2+ T cells were increased in P+ participants after *S. pyogenes* infection (Fig. 3d). No differences were observed in cytotoxic CD8+ T cells, however, CD4+ T cells were decreased 72 h after challenge (Fig. 3d). More detailed analysis of CD4+ T cell subsets revealed reduced frequencies of T-helper type 1 (Th1), Th17, regulatory T cells (Treg) and follicular helper T cells (T$_{FH}$) at 72 h in P+ participants (Fig. 3e), suggesting T cell migration towards the site of infection. Th17 cells are associated with enhanced clearance of *S. pyogenes* at the site of infection, therefore, they might be migrating to infected tissues to mediate local immunity[24].

Unsupervised PCA analysis of the cellular differences between P+ and P− participants at 72 h confirmed that monocytes, dendritic cells, and γδTCR + Vδ2+ T cells contributed to the immune signature of participants who developed pharyngitis (Fig. 3f). Although, there were few differences in immune cell populations at 0 h and 24 h post challenge (Fig. 3), intermediate monocytes (Fig. 3a) and γδTCR + Vδ2+ T cells (Fig. 3d) were higher in P+ participants compared to P− participants prior to challenge. Both intermediate monocytes and γδTCR + Vδ2+ T cells secrete high levels of pro-inflammatory cytokines upon activation and can influence the immune response to infectious disease[25,26]. Collectively, our data support a role for specific subsets of dendritic cells, monocytes, CD4+ T cells and γδTCR + Vδ2+ T cells in acute *S. pyogenes* pharyngitis. Further correlation of these subsets with disease pathogenesis or protection will be a key target for future *S. pyogenes* human-challenge studies and early phase trials evaluating vaccines.

**T cell activation and migration in experimental human *S. pyogenes* pharyngitis.** To further elucidate functional immune responses during acute *S. pyogenes* pharyngitis, we analysed cells for CD69 expression (a marker of activation), CXCR3 (a marker of migration) and intracellular perforin and granzyme-B expression (cytotoxic proteins involved in cell lysis). We found that γδTCR + Vδ2+ T cells and MAIT cells upregulated CD69 at 72 h in P+ participants, but only MAIT cell activation was higher in P+ compared to P− participants at 72 h (Fig. 4a). This was further validated by unsupervised UMAP analysis showing enhanced CD69+ expression on γδTCR + Vδ2+ T cells and MAIT cells in P+ participants (Fig. 4b). In contrast, CD4+ and CD8+ T cells did not upregulate CD69 expression during acute pharyngitis (Fig. 4a). Granzyme-B was also upregulated on γδTCR + Vδ2+ T cells and MAIT cells in P+ participants compared to P− participants (Fig. 4c). While no significant differences were observed in perforin and granzyme-B expression on cytotoxic CD8+ T cells between P+ and P− participants, expression of these cytotoxic molecules increased in CD8+ T cells in P+ participants over time (Fig. 4c). Although *S. pyogenes* is typically considered an extracellular pathogen, it may evade the immune response by invading epithelial cells and macrophages[27,28], and this may account for the differential upregulation of perforin and granzyme-B across T cell subsets.

Expression of CXCR3 was greatly decreased on CD4+ T cells, CD8+ T cells, γδTCR + Vδ2+ T cells and MAIT cells in P+ participants over time (Fig. 4d). Unsupervised UMAP analysis also confirmed the lower CXCR3 expression on γδTCR + Vδ2+ T cells and MAIT cells in participants that developed pharyngitis (Fig. 4e). The decrease in CXCR3 expression (Fig. 4d, e) coincided with increased secretion of serum IP-10 (Fig. 4f), which activates CXCR3 to promote T cell migration[29]. These data suggest a relationship between the chemokine IP-10 and CXCR3 expression which could be important for enhancing T cell migration from the blood.

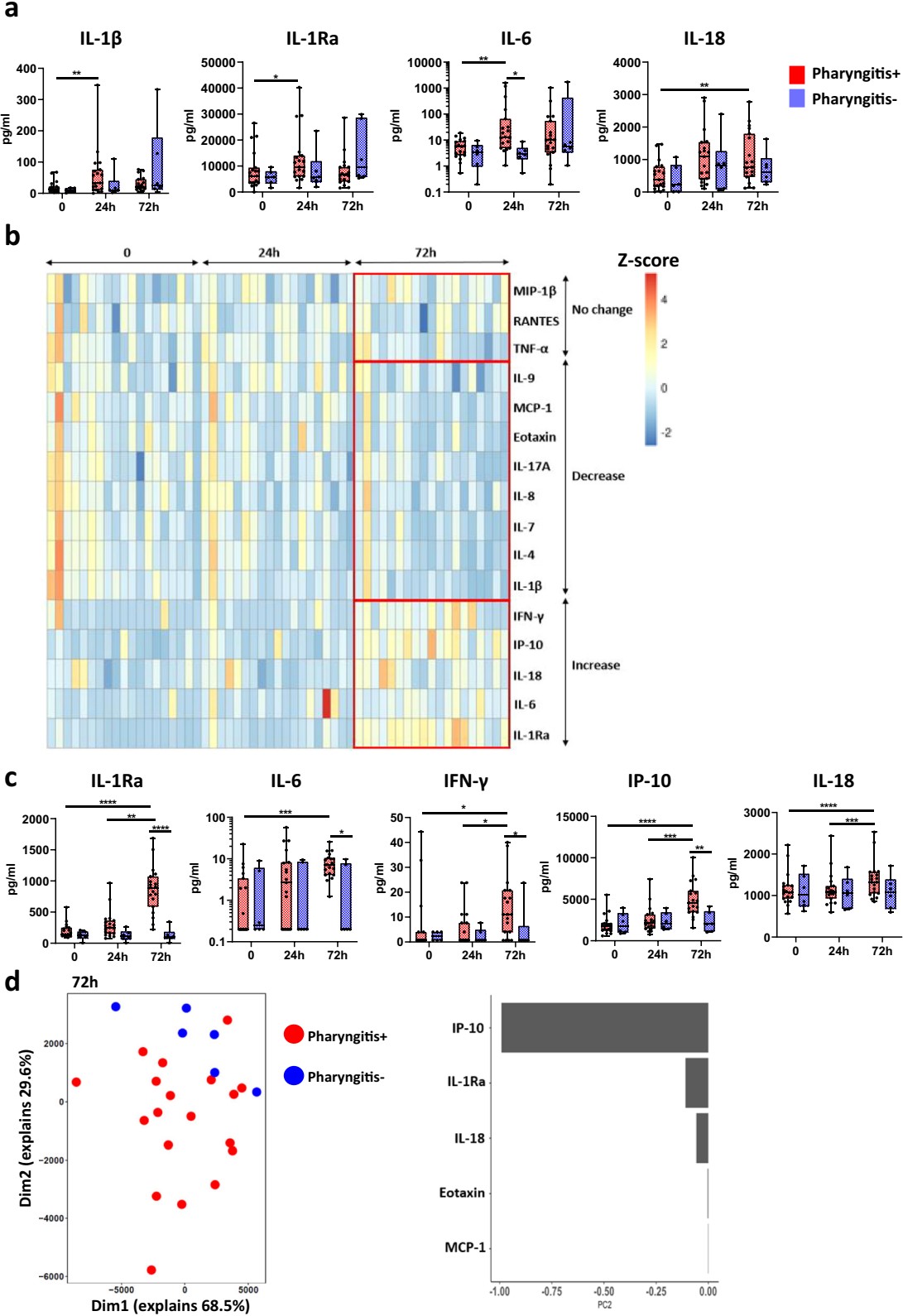

**Fig. 2 Distinct cytokine profiles in participants that developed pharyngitis. a** Saliva cytokines and chemokines at timepoint 0 h (pre-infection) 24 h post-infection (24 h) and 72 h post-infection (72 h) between pharyngitis positive ($n = 19$, red) and negative ($n = 6$, blue) participants. **b** Heatmap summary of serum cytokines and chemokines analysed with normalized Z-score. **c** Serum cytokines and chemokines that had increased during infection. **d** Unsupervised PCA analysis of serum cytokines and chemokines between pharyngitis positive and negative participants at 72 h post-infection and the top 5 factors contributing to differences. Data presented as boxplots show median ± IQR with min–max whiskers. A Friedman's test was used to compare responses over time in P+ and P− groups, while a Mann–Whitney U-test was performed to compare differences between P+ and P− groups. All tests performed were two-tailed and a $p$-value <0.05 was considered significant; *$p < 0.05$, **$p < 0.01$, ***$p < 0.001$, ****$p < 0.0001$.

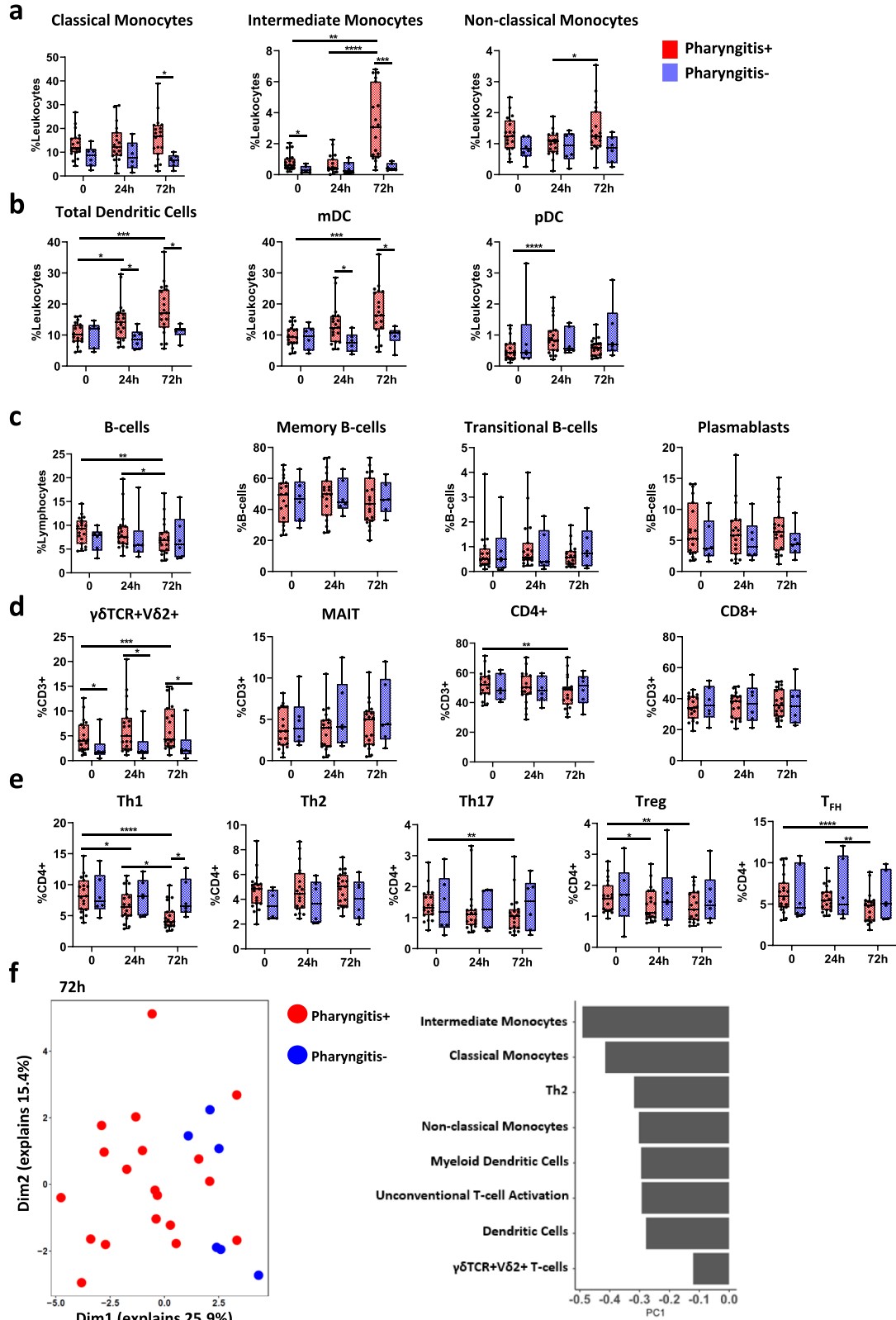

**Fig. 3 Unique cellular profiles in participants that developed pharyngitis. a** Differences in monocyte subsets at timepoint 0 h (pre-infection) 24 h post-infection (24 h) and 72 h post-infection (72 h) between pharyngitis positive ($n = 19$, red) and negative ($n = 6$, blue) participants. **b** Dendritic cell subsets. **c** B cells and B-cell subsets. **d** γδTCR + Vδ2+ T cells, MAIT cells, CD4+ T cells and CD8+ T cells. **e** CD4+ T cell subsets. **f** Unsupervised PCA analysis of cellular subsets between pharyngitis positive and negative individuals at 72 h post-infection and the top 8 factors contributing to differences. Data presented as boxplots show median ± IQR with min−max whiskers. A Friedman's test was used to compare responses over time in P+ and P− groups, while a Mann–Whitney U-test was performed to compare differences between P+ and P− groups. All tests performed were two-tailed and a *p*-value <0.05 was considered significant; *$p < 0.05$, **$p < 0.01$, ***$p < 0.001$, ****$p < 0.0001$.

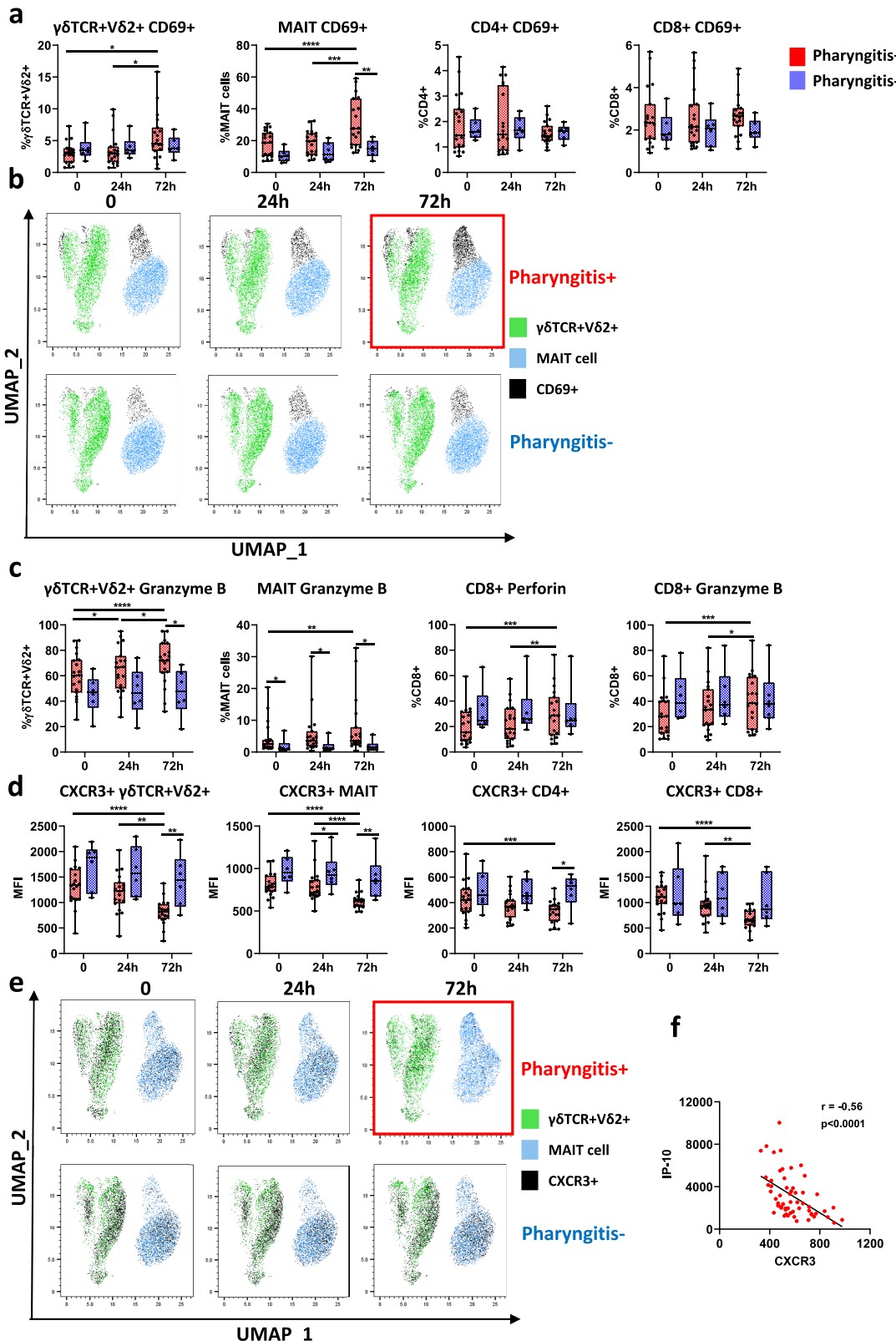

## Discussion

The controlled human-challenge model of *S. pyogenes* infection allowed us to study the immune response to this important human-restricted pathogen, providing greater in-depth analysis than had previously been possible[7]. Here, we have defined a distinct inflammatory immune signature associated with the development of acute *S. pyogenes* pharyngitis in healthy adults.

These findings address a critical knowledge gap in our understanding of host immunity to this pathogen and will help establish a platform for evaluating the immunogenicity and effectiveness of future *S. pyogenes* vaccines.

Acute pharyngitis in participants challenged with *S. pyogenes* was marked by a strong inflammatory immune response mediated by monocytes, dendritic cells and early activation of unconventional

**Fig. 4 Activation and migration of T cell subsets during acute pharyngitis. a** CD69+ activation on T cell subsets timepoint 0 h (pre-infection), 24 h post-infection (24 h) and 72 h post-infection (72 h) between pharyngitis positive ($n = 19$, red) and negative ($n = 6$, blue) participants. **b** Unsupervised UMAP analysis of CD69+ expression (black) on unconventional T cell subsets. **c** Differences in expression of cytotoxic markers (perforin and Granzyme-B) in γδTCR + Vδ2+ T cells, MAIT cells and CD8+ T cells. **d** Changes in T cell CXCR3+ median fluorescence intensity (MFI) over time. **e** Unsupervised UMAP analysis of CXCR3+ expression (black) on unconventional T cell subsets. **f** Spearman's correlation between CXCR3 expression and IP-10 secretion. Data presented as boxplots show median ± IQR with min–max whiskers. A Friedman's test was used to compare responses over time in P+ and P− groups, while a Mann–Whitney U-test was performed to compare differences between P+ and P− groups. All tests performed were two-tailed and a p-value <0.05 was considered significant; *$p < 0.05$, **$p < 0.01$, ***$p < 0.001$, ****$p < 0.0001$.

T cells to initiate the immune response to this pathogen. Monocytes and dendritic cells were increased in the blood of participants who developed pharyngitis. Activation of monocytes to induce pro-inflammatory responses occurs following binding of TLR2 to the Streptococcal Inhibitor of Complement (SIC) protein secreted by *S. pyogenes* which may explain the elevated intermediate monocyte frequency[30]. In addition, the elevation of monocytes and dendritic cells was associated with increased levels of IL-1Ra, IL-18 and IP-10, produced in response to infection[31–33]. These may be important as IP-10 has been previously established as a biomarker for viral and bacterial infections[34–36]. IL-8 and IL-17A facilitate the recruitment of neutrophils, therefore, their reduction in other studies suggests inhibition of neutrophil recruitment during acute pharyngitis[37,38]. *S. pyogenes* employs many strategies to inhibit neutrophil recruitment, one of which is through SpyCEP expression, a protease designed to cleave IL-8[37]. In contrast, IL-4, which was reduced in the serum of P+ participants after infection, has been shown to inhibit neutrophil influx, and the balance between these cytokines may be important in regulating neutrophil activity during *S. pyogenes* infection[39].

The activation of γδTCR + Vδ2+ T cells and MAIT cells, but not other T cell subsets, implies that unconventional T cells may contribute to the inflammatory response during acute *S. pyogenes* pharyngitis. A previous study suggested that MAIT cells were involved in the pathogenesis of human toxic shock syndrome caused by *S. pyogenes*, a more severe manifestation of this bacterial infection[40]. The activation of MAIT cells is likely mediated by the production of IL-18 from innate cells as *S. pyogenes* does not express riboflavin intermediates required to activate MAIT cells via their T-cell receptor[41] and both MAIT cells and γδTCR + Vδ2+ T cells express IL-18R and are potently activated by IL-18[42,43]. T cell migration to the site of infection can be both beneficial and harmful. While T cell migration is required for resolution of disease, excessive migration and release of inflammatory factors can exacerbate disease[44]. The reduced frequency of Treg cells in the blood suggests these cells have migrated to the site of infection to modulate the immune response to *S. pyogenes*, although some evidence suggests that Tregs may facilitate immune evasion by *S. pyogenes*[45]. Moreover, reduced CXCR3 expression observed on T cells during acute pharyngitis suggests that *S. pyogenes* infection promotes migration of CD4+ T cells from the blood. These data suggest that CD4+ T cell migration and activation of unconventional T cells may serve as an early marker of *S. pyogenes* immunity in humans[40,46].

A key focus of vaccine research is to induce protection by emulating natural immune responses. While early antibiotic therapy may shorten the immune response, it is reasonable to presume the pattern of early immune responses in these healthy participants was associated with natural protection. Therefore, vaccine candidates that promote early immune responses by monocytes, dendritic cells and unconventional T cells could enhance natural immunity towards by *S. pyogenes*. There is currently active interest in using adjuvants or other delivery platforms that induce Th1 systemic responses and evidence from animal models show improved protection against *S. pyogenes* disease due to enhanced Th1 immunity[47].

A limitation of this study is the lack of direct sampling at the site of infection. Direct mucosal micro-sampling has contributed significant insight into human-challenge models with other pathogens such as RSV and *S. pneumoniae* and will be explored for use in future *S. pyogenes* human-challenge trials, although there are important safety considerations[12,13]. The strain used in this study was selected for its predictable and limited virulence and the participants being at low risk of complicated disease[16–18], thus analysis of the immune response to other strains of *S. pyogenes* could be beneficial. Although we identified several important features in the human immune response to *S. pyogenes*, increased numbers of participants may strengthen the findings in this study and provide a greater understanding of disease pathogenesis and protective immunity towards *S. pyogenes*. A key strength of this study was the serial collection of blood and saliva samples before and after challenge with *S. pyogenes*, which allowed the identification of changes to the immune system during the critical early stages of infection. Further insights will come from planned randomised placebo-controlled human-challenge trials evaluating *S. pyogenes* vaccines, including a modified sampling schedule informed by these findings, as well as mucosal micro-sampling and whole blood studies that will bring the role of neutrophils into view.

This is the first study to comprehensively characterise the dynamic human cellular immune response to *S. pyogenes* infection. The hallmarks of acute pharyngitis are increased levels of monocyte and dendritic cell subsets, elevation of IL-1Ra, IP-10 and IL-18, activation of unconventional T cells, and migration of B cells and CD4+ T-cell subsets from the blood. This immune signature will now be traced across the *S. pyogenes* research landscape, from in vitro assays, samples from natural human infections, and in future human-challenge and field trials evaluating vaccines. Focused efforts to establish correlates of immune protection and development of vaccines are needed to prevent morbidity and mortality across the *S. pyogenes* clinical spectrum.

## Methods

**Study design and participants**. Strain selection and characterisation, protocol development, and CHIVAS-M75 clinical trial results have previously been described[16–18]. Characterization of the cellular immune response of acute pharyngitis following experimental human challenge with *S. pyogenes* is a secondary outcome of the study protocol[16]. Briefly, healthy adults (aged 20–39) were challenged with *emm75 S. pyogenes* applied directly with a swab to the pharynx, then closely monitored for development of pharyngitis as inpatients for up to 5 nights. Peripheral blood mononuclear cells (PBMCs), serum and saliva samples were collected prior to challenge, 24 and 72 h post-infection or prior to discharge.

The CHIVAS-M75 trial, including immunological studies, was approved by the Alfred Hospital Human Research Ethics Committee (500/17). Written informed consent was obtained from all participants. A safety committee with an independent chair reviewed safety data and clinical details for each participant, meeting at pre-determined timepoints to approve study continuation.

**Study reagents**. RPMI-160, foetal bovine serum (FBS), L-glutamine and penicillin-streptomycin were purchased from Sigma-Aldrich, St. Louis, USA. Anti-mouse compensation beads were purchased from BD Bioscience, San Diego, CA, USA. All flow cytometry antibodies used, and their suppliers, are indicated in Supplementary Tables 1 and 2.

**Flow cytometry**. Cryopreserved PBMCs were thawed at 37 °C then washed with 10 ml $R_{10}$ media (RPMI-1640 medium supplemented with 10% FBS, 2 mM L-glutamine, 1000 IU penicillin-streptomycin) and centrifuged at $400 \times g$ for 5 min. PBMCs were washed with 5 ml PBS and centrifuged at $400 \times g$ for 5 min then blocked (50 μl of 1% human FC-block and 10% normal rat serum in PBS) for 20 min on ice. PBMCs were then washed with 1 ml FACS buffer (PBS supplemented with 2% FBS and 2 mM EDTA) and stained with 50 μl of antibody cocktail 1, 2 or 3 (Supplementary Table 1) for 20 min on ice. PBMCs were washed in 1 ml of FACS buffer then resuspended in 100 μl of fixation buffer (BD, Bioscience, San Diego, CA, USA) and incubated on ice for 20 min. Following, PBMCs were washed twice in permeabilisation buffer prior to intracellular antibody staining. PBMCs were stained for 30 min on ice, washed then resuspended in 100 μl FACS buffer for acquisition using the Cytek Aurora. Compensation was performed at the time of acquisition using compensation beads. Data were analysed using Flowjo v10.7.1 software. UMAP was generated using the UMAP FlowJo plugin and 5000 events per sample were concatenated. Gating strategies are shown in Supplementary Figs. 3, 4, 5.

**Multiplex cytokine/chemokine assay**. A commercial multiplex bead array kit (27-plex human cytokine assay M500KCAF0Y; Bio-Rad, New South Wales, Australia), with the numbers representing the lower limit of detection in pg/ml for each analyte. IL-1β (0.29), IL-1ra (6.21), IL-2 (1.29), IL-4 (0.19), IL-5 (3.63), IL-6 (0.38), IL-7 (1.92), IL-8 (0.85), IL-9 (3.62), IL-10 (1.06), IL-12(p70) (1.43), IL-13 (0.31), IL-15 (12.42), IL-17A (2.44), eotaxin (0.14), FGF-basic (3.26), G-CSF (6.35), GM-CSF (0.48), MCP-1 (0.53), IFN-γ (1.57), TNF-α (3.33), IP-10 (3.41), RANTES (16.72), MIP-1α (0.12), MIP-1β (1.41), PDGF (7.12) and VEGF (18.01) was measured from serum and saliva according to manufacturer's instructions. Results were analysed on a Luminex 200 instrument (Luminex, Texas, USA) fitted with the Bio-Plex Manager Version 6 software and results were reported in pg/ml. IL-18 was measured by a commercial ELISA kit (R&D Systems, Minnesota, USA) according to manufacturer's instructions. Results were measured at an optical density of 450 nm (reference wavelength 630 nm), and concentrations in pg/mL were derived from the standard curve.

**Statistical analysis**. Cellular and cytokine data were presented as a boxplot (median ± IQR with min–max whiskers). A Friedman test was performed to compare cellular and cytokine paired data at baseline (0), 24 h post-infection and 72 h post-infection. An unpaired non-parametric Mann–Whitney U-test was used to compare cellular and cytokine data between P+ and P− groups. A linear regression was used to correlate CXCR3 expression (MFI) and IP-10 concentrations. The data was graphically represented and statistically analysed using Graphpad prism v8 software (Graphpad Software Inc, California, USA). Heatmaps and PCA plots were generated using R software v3.6.1. All tests performed were two-tailed and a $p$-value <0.05 was considered significant.

**Reporting summary**. Further information on research design is available in the Nature Research Reporting Summary linked to this article.

## Data availability

Most data generated or analysed during this study are included in this published article (and its supplementary information files). Any additional data are available from the corresponding authors upon reasonable request. Source data are provided with this paper.

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

## Acknowledgements

J.A. is supported by an Australian Postgraduate Award Scholarship. D.G.P. is supported by CSL Centenary Fellowship. P.V.L. is a recipient of an Australian National Health and Medical Research Council (NHMRC) Career Development Fellowship (GNT1146198). A.C.S. is supported by a Viertel Senior Medical Research Fellowship. B.N. is supported by an NHMRC Investigator Grant (APP1173314). J.O. was supported by an NHMRC postgraduate scholarship (GNT1133299). The challenge study was supported by a NHMRC Project Grant (GNT1099183). We would like to thank Prof. Laura Mackay for her input into the manuscript.

## Author contributions

J.A. collected the data, performed the analyses, and wrote the original draft; S.I. contributed to analysis, visualisation and interpretation of data and critically revised the manuscript; H.R.F., K.I.A., S.J. and B.N. contributed to interpretation of results and revised the manuscript; J.O., A.S., P.V.L. and D.G.P. conceived the study and provided major input into the manuscript revision and data interpretation. All authors approved the final manuscript as submitted and agree to be accountable for all aspects of the work.

## Competing interests

The authors declare no competing interests.
