## [Peer Review File · Nature Communications]

REVIEWER COMMENTS

Reviewer #1 (Remarks to the Author):

General comments

This report describes the inflammatory response in saliva and blood when human volunteers were infected with group A streptococci (GAS). This a bold experiment, which is clearly described. The data are remarkable as the degree of inflammation induced is necessarily very mild, and the samples were not directly obtained from the site of infection. It would be very interesting to extend the model to use micro-sampling of the pharyngeal mucosa but that is for the future and might merit mention in the discussion. The presented methods, analysis, discussion, and conclusions are novel, interesting and relevant to human disease.

Specific comments

Title: clear and explanatory.

Abstract: Clear

Introduction: I would recommend that in addition to the clear description of GAS disease in paragraph 1, some comment on the safety of the model should be made in paragraph 2. How do you know that volunteers will not develop post-infectious GN or other complications?

Results:

a) 2/6 P- volunteers complained of sore throat. They had been inoculated. Some explanation would be reasonable – have other experiments studied the rate of sore throat in sham inoculation?

b) The cytokine profiles are very tough to discuss. The ranges of values measured and the indirect nature of the sample make it very difficult to draw conclusions but the figures are fair in their presentation of the data. The pattern might suggest neutrophil recruitment in this disease – but this is not discussed and curiously IL8 is reduced. Might you mention that? Also the pattern of cytokines with decreased levels in saliva are interesting – both IL4 and IL17 reduced....some thinking about what is being recruited/inhibited would help me as a reader lacking detailed pathway knowledge.

c) At the beginning of the flow cytometry results, I would like to see “peripheral blood” just to be clear. The conclusions in this section are reasonable, but some more comment on whether depletion

indicates tissue compartment recruitment (either current or later in disease) might be helpful. One would expect inflamed cells to recruit to the site of infection, I think?

Discussion

Very reasonable. Line 182 would be nice to clarify “differences” in comparison to what? If you are proposing that this model will aid in developing a future vaccine (line 30), I think it would be reasonable to develop that thinking a little in the discussion. What sorts of vaccine are now suggested, and what might be the limitations given the possibility of immune-mediated disease?

Figures

Figure 1: Unless it is the journal style, I would drop Figure 1 in favour of giving the other figures more space. The content is very clearly and succinctly found in the text.

Figure 2: Saliva not so convincing, but heatmap data very helpful and fig 2c data very helpful. As above, these are tough experiments – a very small infection in a big body, and no direct sampling of the infection site. That can be emphasised perhaps.

Figure 3: Good figure. I’d probably slim the figure to emphasise the main message and take out some of the graphs showing no difference. But the completeness of the data is also an attractive feature of the paper, so I could live with it either way. Certainly it is tough for the reader to go through every graph.

Figure 4: As with other figures, nice elegant analyses and reasonable discussion.

Reviewer #2 (Remarks to the Author):

In Osowicki Lancet Microbe 2021, the authors have previously reported the safety and infection efficiency of challenging 25 healthy adults with *S. pyogenes*. They demonstrated pharyngitis in 85% at the starting dose of $1-3 \times 10^5$ CFU/mL. At the lower dose of $1-3 \times 10^4$ CFU/mL, pharyngitis was diagnosed in one of five participants. They also demonstrated rises in CRP, interferon- γ and

interleukin-6 in blood. In this manuscript, they report the results of their high-dimensional flow cytometric analysis and multiplex cytokine and chemokine assays from serial blood and saliva samples. The data adds incremental value to the original publication but not a new mechanistic understanding of pathogenesis. There are several areas that could be improved in the manuscript:

1) ABSTRACT:

a) The authors make the assumption that the readership will remember from the title that this is a model of streptococcal pharyngitis. This should be clarified.

b) An immune signature is described. It is unclear at what stage in the infection this occurs, where the cellular migration is to or in which compartment the cellular activation occurs.

c) The authors imply that these immune responses are linked to host protection but this evidence is not included.

d) The authors should make clear how these data will assist vaccine and therapeutic evaluation.

2) INTRODUCTION:

a) The burden of disease due *Streptococcus pyogenes* is cited from a 2012 review. Are there any more up to date estimates?

b) The statement “Commonly, *S. pyogenes* infections result in acute pharyngitis” suggests that asymptomatic carriage commonly results in disease. This should be clarified.

2) RESULTS:

A number of the individual results presented do not convincingly substantiate the statements made.

The levels of IL6 in the saliva and blood, and the IFN-gamma in the serum were low. What was the limit of detection for the assay?

The differences in cytokine responses between the P+ and P- in saliva and serum were frequently modest. Although statistically different, it is hard to be sure that the differences are biologically relevant.

The baseline monocytes % tends to be higher in the P+ group than the P- group. This renders the subsequent changes difficult to interpret.

The changes in B-cell % were frequently modest. Although statistically different, the authors have not shown that that the differences are biologically relevant?

The R² value for the relationship between IP10 and CXCR3 is modest.

This section also contains series of unsubstantiated statements:

- a) The authors relate the levels of IP-10 to protective immunity but do not articulate what this consists of or provide evidence of this in their experiments.
- b) The statement “Overall, these data identify changes in chemokine and cytokine expression associated with the development of acute pharyngitis and suggests robust systemic immunity in healthy adults challenged with *S. pyogenes*” implies protective immunity. These are also markers of inflammation which could be harmful.
- c) The authors state “Activation of monocytes to induce pro-inflammatory responses occurs following binding of TLR2 to the Streptococcal Inhibitor of Complement (SIC) protein secreted by *S. pyogenes* which may explain their elevated frequency”. This has not been shown. The statement should be moved to the discussion.
- d) The authors state that “Taken together, our results suggest monocytes and dendritic cells play a key role in experimental human *S. pyogenes* pharyngitis”. However, this causality is not substantiated by the data presented.
- e) The authors state that “Reduced B cells in the blood of P+ participants suggest these cells have migrated to site the infection to initiate antibody responses”. This is speculative and not substantiated by evidence.
- d) The authors state “These data indicate a strong association between the chemokine IP-10 and T-cell migration from the blood” but this has not been demonstrated.

3) DISCUSSION

The authors state that “Acute pharyngitis in participants challenged with *S. pyogenes* was marked by a strong inflammatory immune response mediated by monocytes, dendritic cells and unconventional T-cells, acting in concert to initiate the early immune response to this pathogen”. Co-operation between immune cells has not been demonstrated. That these cells initiate the immune response rather than are a consequence of another immune process has also not been shown.

It is stated that “The early activation of $\gamma\delta$ TCR+V δ 2+ T-cells and MAIT cells, but not other T-cells subsets, implies a role for unconventional T-cells in acute *S. pyogenes* pharyngitis”. This suggests a causal role which is not demonstrated.

The limitations of the data and the model should be discussed.

The authors should make clear how these data will assist vaccine and therapeutic evaluation.

Minor points:

- 1) A reference citation has been included in the abstract.

2) The statistical tests used in the analysis should be made clear in the legends.

Associate Professor Daniel Pellicci,
Murdoch Children's Research Institute
Parkville, VIC 3052, Australia,
dan.pellicci@mcri.edu.au
Wednesday 1st, December 2021

Re: NCOMMS-21-37325-T – Response to Reviewers

We would like to thank both reviewers and convey that their helpful feedback has strengthened the key messages to our manuscript.

Reviewer 1

General comments

This report describes the inflammatory response in saliva and blood when human volunteers were infected with group A streptococci (GAS). This a bold experiment, which is clearly described. The data are remarkable as the degree of inflammation induced is necessarily very mild, and the samples were not directly obtained from the site of infection. It would be very interesting to extend the model to use micro-sampling of the pharyngeal mucosa but that is for the future and might merit mention in the discussion. The presented methods, analysis, discussion, and conclusions are novel, interesting and relevant to human disease.

Specific comments

Title: clear and explanatory.

Abstract: Clear

We thank the reviewer for their very positive comments about our work. We agree that extending this model by micro-sampling the pharyngeal mucosa represents an excellent idea for future human challenge trials with *S. pyogenes*. We now include discussion of this point on page 10, lines 198-200.

Introduction

I would recommend that in addition to the clear description of GAS disease in paragraph 1, some comment on the safety of the model should be made in paragraph 2. How do you know that volunteers will not develop post-infectious GN or other complications?

We thank you for the suggestion, we now include additional discussion regarding the safety of the model in paragraph 3 on (page 4, lines 24-27). Importantly, there were no severe (grade 3) or serious adverse events in participants from this trial. In brief, participants were healthy young adults considered at very low risk of invasive infection or post-infectious complications. Moreover, every participant received antibiotic therapy that succeeded in eradicating carriage of the challenge strain, and no participant developed post-infectious complications during the 6 months follow-up, as measured by urinalysis, electrocardiography, and echocardiography.

Results

a). 2/6 P- volunteers complained of sore throat. They had been inoculated. Some explanation would be reasonable – have other experiments studied the rate of sore throat in sham inoculation?

Sham inoculation has not been studied in this trial. During the challenge procedure, P- participants were subjected to 21 separate throat swabs over 6 days. It is perhaps not too surprising that this was uncomfortable for some, however the overall clinical, microbiological, biochemical, and immunological results for P- participants suggested that they did not exhibit pharyngitis.

For clarity, we have now removed panels b and c from Figure 1 and their related text to avoid confusion surrounding the sore throat symptoms.

b). The cytokine profiles are very tough to discuss. The ranges of values measured and the indirect nature of the sample make it very difficult to draw conclusions but the figures are fair in their presentation of the data. The pattern might suggest neutrophil recruitment in this disease – but this is not discussed and curiously IL8 is reduced. Might you mention that? Also the pattern of cytokines with decreased levels in saliva are interesting – both IL4 and IL17 reduce, some thinking about what is being recruited/inhibited would help me as a reader lacking detailed pathway knowledge.

We appreciate the reviewers concerns regarding the discussion of the cytokine profiles. These data are collected from human blood and saliva samples, thus we anticipate variability from human participants, compared to tightly controlled mouse experiments. We agree that the cytokine pattern might influence neutrophil recruitment in this disease, which could be assessed in future human challenge trials by analysis of whole blood. Of note, *S. pyogenes* employs many strategies to inhibit neutrophil recruitment, where the SpyCEP protein directly cleaves IL-8 during infection and protects against neutrophil-mediated clearance. This might explain the reduced IL-8 levels we observed.

We have now included the following discussion on page 9, lines 162-169:

“IL-8 and IL-17A facilitate the recruitment of neutrophils, therefore, their reduction in other studies suggests inhibition of neutrophil recruitment during acute pharyngitis^{37,38}. *S. pyogenes* employs many strategies to inhibit neutrophil recruitment, one of which is through SpyCEP expression, a protease designed to cleave IL-8³⁷. In contrast, IL-4, which was reduced in the serum of P+ participants after infection, has been shown to inhibit neutrophil influx, and the balance between these cytokines may be important in regulating neutrophil activity during *S. pyogenes* infection³⁹.”

c) At the beginning of the flow cytometry results, I would like to see “peripheral blood” just to be clear. The conclusions in this section are reasonable, but some more comment on whether depletion indicates tissue compartment recruitment (either current or later in disease) might be helpful. One would expect inflamed cells to recruit to the site of infection, I think?

We thank you for the suggestion. We have included peripheral blood mononuclear cells to the beginning of this section on page 6, lines 77-79.

We have also included more in-depth discussion on immune cell recruitment:

“More detailed analysis of CD4+ T-cell subsets revealed reduced frequencies of T-helper type 1 (Th1), Th17, regulatory T-cells (Treg) and follicular helper T-cells (T_{FH}) at 72h in P+ participants (Figure 3e), suggesting T-cell migration towards the site of infection. Th17 cells are associated with enhanced clearance of *S. pyogenes* at the site of infection, therefore, they might be migrating to infected tissues to mediate local immunity”.

Please refer to page 7, lines 101-106, in the tracked changes version of the manuscript.

Discussion

Very reasonable. Line 182 would be nice to clarify “differences” in comparison to what?

Thank you for drawing this to our attention, we now clarify that the statement on line 182 should read “we identified several important features in the human immune response to *S. pyogenes*”, page 10, lines 201-202.

If you are proposing that this model will aid in developing a future vaccine (line 30), I think it would be reasonable to develop that thinking a little in the discussion. What sorts of vaccine are now suggested, and what might be the limitations given the possibility of immune-mediated disease?

This point was raised by both reviewers and we thank them for the suggestion. Given that an *S. pyogenes* vaccine has not been evaluated for efficacy in humans against any *S. pyogenes* clinical syndrome in over 40 years, our model will be used for future randomised controlled trials to evaluate candidate vaccines for protection against experimental human pharyngitis. Currently active vaccine programs include multivalent and multicomponent peptide and protein conjugate vaccine products, employing type-specific and conserved antigens from *S. pyogenes*. Post-infectious complications following *S. pyogenes* infections are rare, especially serious complications of acute rheumatic fever and rheumatic heart disease among healthy young adults such as the participants in our study.

We now include the following discussion on pages 9-10, lines 187-194:

“A key focus of vaccine research is to induce protection by emulating natural immune responses. While early antibiotic therapy may shorten the immune response, it is reasonable to presume the pattern of early immune responses in these healthy participants was associated with natural protection. Therefore, vaccine candidates that promote early immune responses by monocytes, dendritic cells and unconventional T-cells, could enhance natural immunity towards *S. pyogenes*. There is currently active interest in using adjuvants or other delivery platforms that induce Th1 systemic responses and evidence from animal models show improved protection against *S. pyogenes* disease due to enhanced Th1 immunity⁴⁷”

6. Figure 1: Unless it is the journal style, I would drop Figure 1 in favour of giving the other figures more space. The content is very clearly and succinctly found in the text.

We thank the reviewer for this suggestion and the positive feedback. We are in favour of keeping most of figure 1 as it helps to set the scene for the data that follows, although we now have removed panels b and c as they are less relevant to the present study.

7. Figure 2: Saliva not so convincing, but heatmap data very helpful and fig 2c data very helpful. As above, these are tough experiments – a very small infection in a big body, and no direct sampling of the infection site. That can be emphasised perhaps.

Thank you for the comment. As discussed above and mentioned on page 10, lines 195-198, we agree that direct micro-sampling of the pharyngeal mucosa would be very useful for future challenge trials.

8. Figure 3: Good figure. I'd probably slim the figure to emphasise the main message and take out some of the graphs showing no difference. But the completeness of the data is also an attractive feature of the paper, so I could live with it either way. Certainly, it is tough for the reader to go through every graph.

Thank you for the suggestion, we are happy to be guided by the editorial team, although where possible, we would prefer to provide the reader a comprehensive understanding of cellular phenotypes in response to *S. pyogenes* infection.

9. Figure 4: As with other figures, nice elegant analyses and reasonable discussion.

Thank you for the comment.

Reviewer 2

General comments

In Osowicki Lancet Microbe 2021, the authors have previously reported the safety and infection efficiency of challenging 25 healthy adults with *S. pyogenes*. They demonstrated pharyngitis in 85% at the starting dose of $1-3 \times 10^5$ CFU/mL. At the lower dose of $1-3 \times 10^4$ CFU/mL, pharyngitis was diagnosed in one of five participants. They also demonstrated rises in CRP, interferon- γ and interleukin-6 in blood. In this manuscript, they report the results of their high-dimensional flow cytometric analysis and multiplex cytokine and chemokine assays from serial blood and saliva samples. The data adds incremental value to the original publication but not a new mechanistic understanding of pathogenesis. There are several areas that could be improved in the manuscript:

This is the first human challenge model with *S. pyogenes* in almost 5 decades - the additional cytokine and chemokine analysis and comprehensive immune profiling using high-dimensional flow cytometry represent a significant advance in our understanding of the human immune response to this important pathogen. Our work describes many novel findings that will be invaluable for future human challenge and vaccine trial studies.

1) ABSTRACT:

a) The authors make the assumption that the readership will remember from the title that this is a model of streptococcal pharyngitis. This should be clarified.

Thank you for suggestion. We have now re-iterated throughout the manuscript that this is an acute streptococcal pharyngitis human model.

b) An immune signature is described. It is unclear at what stage in the infection this occurs, where the cellular migration is to or in which compartment the cellular activation occurs.

Thank you for this comment. We have removed the suggestion of cellular migration from the abstract and left this to the results and discussion. The text on page 3 now reads:

“reduced circulation of B-cells and CD4⁺ T-cell subsets (Th1, Th17, Treg, T_{FH}) during the acute phase”.

c) The authors imply that these immune responses are linked to host protection but this evidence is not included.

Thank you for the comment. We have softened our language to be less conclusive about whether the immune response is protective, but we would like to highlight that the challenge was performed in healthy adults and therefore likely to be self-limiting and resolved in the absence of antibiotic treatment. We include further discussion of this on pages 9-10, lines 187-194.

d) The authors should make clear how these data will assist vaccine and therapeutic evaluation.

Thank you for this suggestion, which was shared by reviewer 1. Please refer to our responses above. We have re-worded our statement to clarify how these data will be beneficial:

“These data will advance research to establish correlates of immune protection and focus the evaluation of vaccines. This is urgently needed to address the unmet public health burden of uncontrolled disease caused by this pathogen”. Please refer to page 3 in the manuscript.

We also provide increased discussion of this point on pages 9-10, lines 187-194.

2) INTRODUCTION:

a. The burden of disease due *Streptococcus pyogenes* is cited from a 2012 review. Are there any more up to date estimates?

Thank you for the suggestion. Since 2012, there have been a number of publications estimating the global burden of disease for different clinical syndromes caused by *S. pyogenes*, although there has been no single authoritative update of estimates for all *S. pyogenes* syndromes (a new full review is underway currently through an international collaboration, but the data are not published yet). We have included an additional citation that is more recent.

b. The statement “Commonly, *S. pyogenes* infections result in acute pharyngitis” suggests that asymptomatic carriage commonly results in disease. This should be clarified.

We have reworded this to sentence to read “Acute pharyngitis is the most common of all clinical syndromes due to *S. pyogenes*”. Please refer to page 4, lines 4-5.

3) RESULTS:

A number of the individual results presented do not convincingly substantiate the statements made. The levels of IL6 in the saliva and blood, and the IFN-gamma in the serum were low. What was the limit of detection for the assay?

Thank you for this comment. We agree that expression of some analytes such as IL-6 and IFN γ were low in some samples, although we only discuss cytokine/chemokine data that reached significance. We have stated in our methods that we used a Luminex assay and for IL-6 and IFN γ , the limit of detection was 0.42-6,918 pg/ml and 0.6-9,820 pg/ml, respectively.

The differences in cytokine responses between the P+ and P- in saliva and serum were frequently modest. Although statistically different, it is hard to be sure that the differences are biologically relevant.

We agree with the reviewer that some of the differences between the P+ and P- groups are modest, although the main objective of this study was to evaluate how the immune response changed in those that developed acute pharyngitis. The difference in cytokine/chemokine responses over time from pre-infection (0h) to acute pharyngitis diagnosis (72h) was often very large for some cytokines. While our data suggests these cytokines may have an important role during *S. pyogenes* infection, we are unable to provide conclusive evidence of their role in protection or pathogenesis, and thus many of our comments are suggestive.

The baseline monocytes % tends to be higher in the P+ group than the P- group. This renders the subsequent changes difficult to interpret.

Thank you for this comment. We agree that monocyte populations were generally higher in the P+ group, which might lead to a more robust response to infection. This is speculative, so we have refrained from making any statements to suggest this. We have re-worded the sentence describing these differences in the results to make it clear we are focussing on intermediate monocytes which had a very strong response, page 8, lines 157-159. We have also moved this section to the discussion as suggested by the reviewer in the comment below. This now reads "Activation of monocytes to induce pro-inflammatory responses occurs following binding of TLR2 to the Streptococcal Inhibitor of Complement (SIC) protein secreted by *S. pyogenes* which may explain the elevated intermediate monocyte frequenc³⁰".

The changes in B-cell % were frequently modest. Although statistically different, the authors have not shown that that the differences are biologically relevant?

We have placed emphasis on the fact that B-cell frequencies decreased only in those who developed pharyngitis. This suggests that these cells are responding to infection, and we have postulated that they may be migrating to the site of infection. Subsequent studies are needed to provide direct evidence for the biological relevance of B cells during *S. pyogenes* infection. Thus, we believe our discussion is reasonable.

The R2 value for the relationship between IP10 and CXCR3 is modest.

Thank you for this comment. The correlation should be represented by the r-value since this measures the strength and direction of the relationship. The r-value is -0.56 which is indicative of a moderate correlation between these two variables. Please refer to figure 4f for these changes.

This section also contains series of unsubstantiated statements:

We thank the reviewer for their comments, which we address below. As this trial was performed in humans, it is difficult to provide conclusive evidence. However, we agree that the language should be softened accordingly.

a) The authors relate the levels of IP-10 to protective immunity but do not articulate what this consists of or provide evidence of this in their experiments.

Thank you for this comment. We have softened the language to now state “IP-10 can directly kill *S. pyogenes* which suggests that the elevated IP-10 seen in our study may be related to protective immunity by enhancing bacterial clearance²⁰”.

Please refer to page 6, lines 66-67, in the tracked changes version of the manuscript.

b) The statement “Overall, these data identify changes in chemokine and cytokine expression associated with the development of acute pharyngitis and suggests robust systemic immunity in healthy adults challenged with *S. pyogenes*” implies protective immunity. These are also markers of inflammation which could be harmful.

Yes, we agree that these markers might be harmful, although assert our trial was performed in healthy adult participants who are less likely to develop severe disease. To provide a more balanced view of this statement, we have modified the statement as follows:

“Overall, these data identify changes in chemokine and cytokine expression associated with the development of acute pharyngitis”.

Please refer to page 6, lines 74-75, in the tracked changes version of the manuscript.

c) The authors state “Activation of monocytes to induce pro-inflammatory responses occurs following binding of TLR2 to the Streptococcal Inhibitor of Complement (SIC) protein secreted by *S. pyogenes* which may explain their elevated frequency”. This has not been shown. The statement should be moved to the discussion.

Thank you for the suggestion. We have now moved this statement to the discussion.

Please refer to page 8, lines 157-159, in the tracked changes version of the manuscript.

d) The authors state that “Taken together, our results suggest monocytes and dendritic cells play a key role in experimental human *S. pyogenes* pharyngitis”. However, this causality is not substantiated by the data presented.

The role of monocytes and dendritic cells have previously been shown in the literature to be important innate immune cells required for phagocytosis, neutrophil activation and T cell activation in response to *S. pyogenes*. While it is difficult to demonstrate causality in our study, we have included additional discussion of this to support our statement.

“In response to *S. pyogenes* monocytes and dendritic cells are crucial to phagocytosis, pro-inflammatory cytokine secretion, neutrophil recruitment, and T-cell activation^{7,21}. In mice, depletion of myeloid dendritic cells resulted in uncontrolled dissemination of *S. pyogenes*²²”.

Please refer to page 6, lines 86-89 in the manuscript.

e) The authors state that “Reduced B cells in the blood of P+ participants suggest these cells have migrated to site the infection to initiate antibody responses”. This is speculative and not substantiated by evidence.

Yes, we agree this is speculative, which is why we suggest that these cells have migrated to the site of infection. However, we have removed the suggestion that they have migrated to initiate antibody responses.

The sentence now reads: “B-cells decreased in P+ participants suggesting migration to the site of infection”.

Please refer to page 6, line 92 in the manuscript.

d) The authors state “These data indicate a strong association between the chemokine IP-10 and T-cell migration from the blood” but this has not been demonstrated.

Thank you for this comment. We have re-worded this sentence as follows:

“These data suggest a relationship between the chemokine IP-10 and CXCR3 expression which could be important for enhancing T-cell migration from the blood”

Please refer to page 8, lines 142-144, in the manuscript.

4) DISCUSSION

The authors state that “Acute pharyngitis in participants challenged with *S. pyogenes* was marked by a strong inflammatory immune response mediated by monocytes, dendritic cells and unconventional T-cells, acting in concert to initiate the early immune response to this pathogen”. Co-operation between immune cells has not been demonstrated. That these cells initiate the immune response rather than are a consequence of another immune process has also not been shown.

Thank you for this comment. While we agree with the reviewer that co-operation between immune cells has not been clearly demonstrated, the evidence suggests that this may be likely. Unconventional T cells express high levels of IL-18R and are known to be responsive to IL-18, which we show is increased in P+ participants in the saliva and blood. We also demonstrate activation of unconventional T cells as measured by CD69 expression. This provides evidence for the role of unconventional T cells in acute pharyngitis in participants challenged with *S. pyogenes*. We have re-worded the sentence as follows:

“Acute pharyngitis in participants challenged with *S. pyogenes* was marked by a strong inflammatory immune response mediated by monocytes, dendritic cells and early activation of unconventional T-cells to initiate the immune response to this pathogen”.

Please refer to page 8, lines 153-155, in the manuscript.

It is stated that “The early activation of $\gamma\delta$ TCR+V δ 2+ T-cells and MAIT cells, but not other T-cells subsets, implies a role for unconventional T-cells in acute *S. pyogenes* pharyngitis”. This suggests a causal role which is not demonstrated.

Please refer to our response above. We have now modified the sentence to read “The activation of $\gamma\delta$ TCR+V δ 2+ T-cells and MAIT cells, but not other T-cell subsets, implies that unconventional T-cells may contribute to the inflammatory response during acute *S. pyogenes* pharyngitis”.

Please refer to page 9, lines 170-172, in the manuscript.

The limitations of the data and the model should be discussed.

Thank you for this suggestion. Limitations of our study were included, although we acknowledge and appreciate other limitations to the model and data raised by both reviewers. We have now included more in-depth discussion of potential limitations on page 10, lines 195-204.

“A limitation of this study is the lack of direct sampling at the site of infection. Direct mucosal micro-sampling has contributed significant insight into human challenge models with other pathogens such as RSV and *S. pneumoniae* and will be explored for use in future *S. pyogenes* human challenge trials, although there are important safety considerations^{12,13}. The strain used in this study was selected for its predictable and limited virulence and the participants being at low risk of complicated disease¹⁶⁻¹⁸, thus analysis of the immune response to other strains of *S. pyogenes* could be beneficial. Although we identified several important features in the human immune response to *S. pyogenes*, increased numbers of participants may strengthen the findings in this study and provide a greater understanding of disease pathogenesis and protective immunity towards *S. pyogenes*”.

The authors should make clear how these data will assist vaccine and therapeutic evaluation.

Thank you for the suggestion, which was also shared by reviewer 1. We provide discussion of this point on pages 9-10, lines 187-194.

MINOR POINTS:

1) A reference citation has been included in the abstract.

Thank you, we have now removed this.

2) The statistical tests used in the analysis should be made clear in the legends.

Thank you for this suggestion, we have updated the figure legends to include the analyses used. Please refer to all figure legends and supplementary figure legends 1, 2 and 6 for these changes.

REVIEWERS' COMMENTS

Reviewer #1 (Remarks to the Author):

The authors have now addressed my concerns and made every effort to respond to suggestions.

Thank you.

Reviewer #2 (Remarks to the Author):

The authors have satisfactorily addressed all the comments made by the reviewers except that they have not put the IL-6 and IFN γ limits of detection in the methods as stated in the response letter (unless I have missed it).

Associate Professor Daniel Pellicci,
Murdoch Children's Research Institute
Parkville, VIC 3052, Australia,
dan.pellicci@mcri.edu.au
Monday 17th, January 2022

Re: NCOMMS-21-37325-A – Response to Reviewers

We would like to thank both reviewers and convey that their helpful feedback.

Reviewer 1

General comments

The authors have now addressed my concerns and made every effort to respond to suggestions

Thank you.

Reviewer 2

General comments

The authors have satisfactorily addressed all the comments made by the reviewers except that they have not put the IL-6 and IFN γ limits of detection in the methods as stated in the response letter (unless I have missed it).

Thank you. We apologise for the oversight. The limit of detection is now included in the methods for each analyte on page 16, lines 370-377.

Kind Regards,

A/Prof Daniel Pellicci